# Genome-Wide Analysis of the Soybean Calmodulin-Binding Protein 60 Family and Identification of GmCBP60A-1 Responses to Drought and Salt Stresses

**DOI:** 10.3390/ijms222413501

**Published:** 2021-12-16

**Authors:** Qian Yu, Ya-Li Liu, Guo-Zhong Sun, Yuan-Xia Liu, Jun Chen, Yong-Bin Zhou, Ming Chen, You-Zhi Ma, Zhao-Shi Xu, Jin-Hao Lan

**Affiliations:** 1College of Agronomy, Qingdao Agricultural University, Qingdao 266109, China; yu2020kkk@163.com (Q.Y.); liuyali5699@126.com (Y.-L.L.); yuanxialiu@163.com (Y.-X.L.); 2Institute of Crop Sciences, Chinese Academy of Agricultural Sciences (CAAS)/National Key Facility for Crop Gene Resources and Genetic Improvement, Key Laboratory of Biology and Genetic Improvement of Triticeae Crops, Ministry of Agriculture, Beijing 100081, China; sunguozhong@caas.cn (G.-Z.S.); chenjun01@caas.cn (J.C.); zhouyongbin@caas.cn (Y.-B.Z.); chenming02@caas.cn (M.C.); mayouzhi@caas.cn (Y.-Z.M.)

**Keywords:** drought tolerance, salt tolerance, CBP60 proteins, hairy root assay, soybean, *Arabidopsis*

## Abstract

Calmodulin-binding protein 60 (CBP60) members constitute a plant-specific protein family that plays an important role in plant growth and development. In the soybean genome, nineteen CBP60 members were identified and analyzed for their corresponding sequences and structures to explore their functions. Among GmCBP60A-1, which primarily locates in the cytomembrane, was significantly induced by drought and salt stresses. The overexpression of *GmCBP60A-1* enhanced drought and salt tolerance in *Arabidopsis*, which showed better state in the germination of seeds and the root growth of seedlings. In the soybean hairy roots experiment, the overexpression of *GmCBP60A-1* increased proline content, lowered water loss rate and malondialdehyde (MDA) content, all of which likely enhanced the drought and salt tolerance of soybean seedlings. Under stress conditions, drought and salt response-related genes showed significant differences in expression in hairy root soybean plants of *GmCBP60A-1*-overexpressing and hairy root soybean plants of RNAi. The present study identified *GmCBP60A-1* as an important gene in response to salt and drought stresses based on the functional analysis of this gene and its potential underlying mechanisms in soybean stress-tolerance.

## 1. Introduction

During growth and development, plants face various biotic stresses, such as pathogen infection and herbivore attack, and abiotic stresses, including drought, heat, cold and salt [1]. In response to abiotic stresses, various signaling pathways were activated to promote the plants sensing of the stimuli and then to trigger the adequate cellular responses, which include calcium signaling pathways [2], MAPK (mitogen-activated protein kinases) cascade and other signaling pathways [3,4]. Among them, the calcium signaling pathway appears to be particularly important because almost all physiological activities are regulated by Ca^2+^ in eukaryotes [5,6,7].

The Ca^2+^ was first known as a necessary nutrient for plant growth [8], many modern studies have shown that it is a ubiquitous secondary messenger that mediates stimuli-response in the regulation of diverse cellular functions [9]. Large amounts of Ca^2+^ in the intracellular and extracellular environments flow in and out of cells through channels in the cytomembrane, such as cation/proton exchangers (CAXs), glutamate receptor-like channels (GLRs), two-pore channels (TPCs), cyclic nucleotide-gated channels (CNGCs), ER (endoplasmic reticulum)-type Ca^2+^-ATPases (ECAs) and auto-inhibited Ca^2+^-ATPases (ACAs) [10,11,12,13]. Cytoplasmic Ca^2+^ transduces a variety of regulatory information based on oscillations in its concentration [14,15]. This signal transmission method is called calcium oscillation. Previous studies suggest that a change of cytoplasmic Ca^2+^ concentration is usually induced by plant responses to a variety of biotic and abiotic stimuli [2]. The variations of cytoplasmic Ca^2+^ concentration is detected by various sensor proteins, such as calmodulins (CaMs), CaM-like proteins (CMLs), calcium-dependent protein kinases (CDPKs) and calcineurin B-like proteins (CBLs) in the plant cell [16]. Calcium ions (Ca^2+^) are crucial in the activation of stress-related signaling cascades [17].

As one of the most extensively studied Ca^2+^-sensing proteins, CaM has been shown to be involved in the transduction of the Ca^2+^-mediated signal pathway [18,19]. This calcium-binding protein consists of two globular domains, each with two Ca^2+^-binding EF-hand motifs [20]. CaM responds to the change of Ca^2+^ concentration and transmits the signal to the target proteins [21]. Upon binding to Ca^2+^, the hydrophobic surfaces in each globular domain are exposed. The exposed globular domains are called the CaM-binding domain (CBD) which then interacts with the characteristic amphiphilic structure of CaM-binding proteins (CBPs) [21]. CaM can regulate the development of plants and stimuli-response by interacting with various CBPs [22]. Thus, as a downstream target protein of Ca^2+^-CaM, CBPs participate in many of life’s biological processes that involve Ca^2+^-CaM [22]. The CBP family mainly includes calmodulin-binding transcription activators (CAMTAs), WRKYs, MYBs, NACs and the CaM-binding protein family (CBP60s) [5,23].

CBP60 family had reported contains eight different members in *Arabidopsis* encoded by the genes *AtCBP60*‘*a*’ to ‘*g*’ and *AtSARD1* (Systemic Acquired Resistance Deficient 1) [24]. On the one hand, all these proteins have been shown to bind DNA through a highly conserved central region and regulate the expression of specific genes in *Arabidopsis* [25]; on the other hand, previous studies suggest that the CBP60 family likely have binding CaM and Ca^2+^ by conserved domain to participate in Ca^2+^-mediated signal pathway in *Arabidopsis* [6,7,26]. As plant-specific protein family, CBP60 family plays an important role when plant facing various stresses. Most of the functions of CBP60 family members are related to disease resistance, immunity, and SA accumulation [27,28]. Among, *AtCBP60a*, *AtCBP60g* and *AtSARD1* have been implicated in plant defense responses and the accumulation of salicylic acid (SA). CBP60g was also reported it repressed anthocyanin accumulation induced by sucrose and kinetin [29]. However, their functions in response to abiotic stress remain largely elusive. Despite the lack of such studies, they are not entirely absent. Previous study have shown that *AtCBP60g* enhanced the sensitivity of ABA as a positive regulator for drought tolerance in *Arabidopsis* [30] and *AtCBP60f* and *AtCBP60g* are upregulated in response to brassinosteroid (BR) and salt treatments [31].

Soybean (*Glycine max*) is one of the most economically and nutritionally crucial crops in the world [32]. However, soybean production is threatened by drought and salt stresses, as well as other stresses. Therefore, to facilitate the continued expansion of land areas for soybean cultivation, we need to improve soybean’s drought and salt tolerances to improve soybean productivity and security in less-than-ideal lands [33]. In our study, we endeavored to characterize the GmCBP60 family members by investigating the phylogenetic relationships, domain structures, chromosomal localizations, and expression patterns of GmCBP60s in response to abiotic stresses. Our findings will be useful resources for future studies to unravel the functions of the *GmCBP60* genes and will contribute to our understanding of the evolutionary history of the *CBP60* genes in different species. Furthermore, we focused on exploring the functional mechanisms of a specific soybean gene, *GmCBP60A-1*, in response to salt and drought treatments. Collectively, the current research provided insights for the future functional study of *GmCBP60* genes and may be valuable for soybean breeding.

## 2. Results

### 2.1. Relationships of CBP60 Members in Various Species

In the EnsemblPlants database, 19 CBP60 proteins were identified in the soybean genome. To examine the phylogenetic relationships among the calmodulin binding domain proteins in soybean and other species, an unrooted tree was constructed using the maximum likelihood method (ML). The phylogenetic tree had three evolutionary branches, designated A, B and C, composed of 19 rice, 8 *Arabidopsis*, 18 maize and 19 soybean (Figure 1). The three branches showed the evolutionary relationship of homologous genes. The number of CBP60s in each species was variable within the branches. The genes distributed on branch A and B were most from maize and rice. Compared to rice in dicot, *CBP60* genes in soybean showed a closer relationship with that in maize because they always clustered closely to each other in the phylogenetic tree. Furthermore, branch A did not have any soybean genes. Thus, the ancestral gene of branch A appeared to be lost in the soybean lineage.

### 2.2. Sequences Analysis of GmCBP60 Family Members

We obtained the sequences of GmCBP60 members from the EnsemblPlants website. The GmCBP60 genes chromosomal positions were depicted based on the gene physical location information of the soybean genome. Nineteen GmCBP60 family members were distributed on 10 of 20 soybean chromosomes, none of them was found on chromosomes 1, 2, 4, 6, 11, 12, 16, 18 and 20. Chromosome 3 contained the largest number of GmCBP60 genes (Figure 2A). *Cis*-regulatory events were complex processes that involved chromatin accessibility, transcription factor binding, DNA methylation, histone modifications, and the interactions between them. In addition, the promoters of GmCBP60s (Figure 2B) contained a wealth of stress-related *cis-acting* elements, which suggested that GmCBP60s might function in a variety of physiological and biochemical processes in plants. Among this, only 6 members have both ABAR (ABA-responsive element) and DER (Dehydration-responsive element) *cis-acting* elements (Table 1), which were related to abiotic stress. According to the sequences of their encoding proteins, all members of the GmCBP60s in soybean had a CaM-binding domain (Figure 2C). Ten conserved motifs were detected in (Figure 2D). Among them, the motif 1, 2, 3, 5 and 10, were widely distributed in all family members (Appendix A).

### 2.3. Expression of GmCBP60s in Different Treatments

In the de novo transcriptome sequencing data of soybean following drought and salt treatments [34], it was found that a number of *GmCBP60* genes were triggered by drought and salt stimuli (Figure 3A and Appendix A). According to gene expression induced by stresses, we screened out seven genes from GmCBP60s family, including *GmCBP60A-1*, *GmCBP60B-2*, *GmCBP60B-3*, *GmCBP60B-4*, *GmCBP60D-1*, *GmSARD1-1*, *GmSARD1-like*. Subsequently, we detected the response patterns of the seven genes to salt (Figure 3B), drought (Figure 3C) and PEG (Figure 3D) treatments by RT-qPCR. The results showed that *GmCBP60A-1*, *GmSARD1-1* and *GmSARD1-like* transcripts were strongly responsive to salt stress compared with others. Meanwhile, *GmCBP60A-1* displayed the greatest induction with drought and PEG treatments. In addition, *GmCBP60A-1* was expressed in various tissues of soybean (Appendix A), indicating that *GmCBP60A-1* might be important for the regulation of plant growth and development compared with *GmSARD1-1* and *GmSARD1-like*. Thus, we focused on *GmCBP60A-1* in our subsequent analyses.

### 2.4. Localtion of GmCBP60A-1

The protein localization is usually related to its function, we used the method of transient transformation of protoplasts to study preliminarily the localization of GmCBP60A-1. The GmCBP60A-1-GFP fluorescence signal was observed in the cytomembrane (Figure 4A,B). To further confirm the localization of GmCBP60A-1, the proteins of the plant nucleus, cytoplasm and cytomembrane were isolated by using Minute^TM^ Plasma Membrane Protein Isolation Kit (Figure 4C). The result proved that GmCBP60A-1 existed in the cytomembrane and cytoplasm, but it was primarily localized in the cytomembrane.

### 2.5. Effect of Salt and Drought Conditions on Transgenic Arabidopsis

To study the role of the *GmCBP60A-1* gene in the heterologous model system under drought and salt stresses, the transgenic *Arabidopsis* lines were constructed. We tested the effect of PEG6000 and NaCl on the germination of transgenic *Arabidopsis* seeds. There was no difference in seed germination between the wild-type and transgenic plants under normal conditions (Figure 5A). In the presence of PEG6000 and NaCl treatments of different concentrations, seed germination was significantly suppressed in both overexpression (OE) and wild-type (WT) lines, but the inhibition of OE seeds germination was much less than WT plants. When treated with 75 mM NaCl for 36 h, the germination rate of wild-type seeds was 44%, while that of transgenic lines was 66–77%. When treated with 6% PEG6000 for 36 h, the germination rate of wild-type seeds was 70%, while that of transgenic lines was 87–91% (Figure 5B).

Differential inhibition of root growth between WT and transgenic seedlings was also observed with drought and salt treatments. Five-day-old seedlings under normal conditions were transferred to 1/2 MS medium with PEG6000 or NaCl of different concentrations. After one week, roots lengths of WT plants and transgenic plants were inhibited significantly, but OE plants were inhibited to a lesser extent (Figure 6A). The root length of WT plants reached 4.56 cm compared with 5.87 to 6.44 cm for that of transgenic lines treated with 75 mM NaCl, and the root length of WT plants reached 2.51 cm compared with 4.45 to 5.59 cm for transgenic lines treated with 6% PEG6000 (Figure 6B,C).

### 2.6. Effect of Salt Conditions on Transgenic Soybean Hairy Roots

Three transgenic soybean plants OE (the hairy root plants were infected *GmCBP60A-1*-OE), EV (the hairy root plants were infected pCAMBIA3301 vector) and RNAi (the hairy root plants were infected *GmCBP60A-1*-RNAi) were generated by *A. rhizogenes*-mediated hairy roots transformation (Figure 7A) to investigate the function of *GmCBP60A-1* in soybean. The growth of different transgenic hairy roots soybean at V3 stage was similar under normal conditions (Figure 7A). Differential inhibition of plants growth between three transgenic hairy roots soybean seedlings was also observed following treatment with salt. In the presence of salt for three days, hairy roots plants displayed significantly wilting in both RNAi and EV. The hairy roots plants of EV showed degree of leaf yellowing at a lesser extent than RNAi hairy roots plants; at the same condition, growth of OE hairy roots seedlings was not wilt (Figure 7A). After nine days of salt treatment, the growth of hairy roots plants showed significantly wilting in both EV and OE plants, but wilt extent of EV plants growth was much more obvious than OE plants; and growth of RNAi plants was nearly dead (Figure 7A).

RNAi plants displayed wilting after irrigated NaCl solution for 3 days, but less extreme wilting in EV plants and a near-healthy appearance in OE plants (Figure 7A). Under salt stress, there were much lower levels of chlorophyll (Figure 7E) and proline (Figure 7H) and much higher levels of NOX activity (Figure 7G), MDA (Figure 7K), H_2_O_2_ (Figure 7I) and O^2−^ (Figure 7J) in RNAi plants relative to EV plants, and opposite effects for OE plants. The assays of leaf staining could help explain that RNAi plants contained more harmful products compared to EV plants, and in turn OE plants had less noxious substances than EV plants (Figure 7B–D). In a word, *GmCBP60A-1* has a positive effect on hairy root soybeans under salt treatment.

### 2.7. Effect of Drought Conditions on Transgenic Soybean Hairy Roots

To determine whether drought tolerance of soybeans was affected by *GmCBP60A-1*, different transgenic hairy roots soybean plants at V3 stage were without holding water to give plants drought stress. The growth of different plants was similar under normal conditions (Figure 8A). Differential inhibition of plants growth between three plants seedlings was also observed following treatment with drought. In the presence of drought for 13 days, plants growth displayed significantly wilting in both RNAi and EV plants, but the wilt extent of RNAi plants was much more obvious than EV plants; at the same condition, growth of OE seedlings were not wilt (Figure 8A). After rewatering for three days, OE plants grew new leaves and the recovery situation was better than that of EV plants, whereas all RNAi plants were unable to recover (Figure 8A).

Under drought stress, there were much lower levels of chlorophyll (Figure 8E) and proline (Figure 8H) and much higher levels of NOX activity (Figure 8G), MDA (Figure 8K), H_2_O_2_ (Figure 8I) and O^2−^ (Figure 8J) in RNAi plants relative to EV plants, and opposite effects for OE plants. The assays of leaf staining could help explain that RNAi plants contained more harmful products than EV plants, contrary to OE plants (Figure 8B–D). In short, *GmCBP60A-1* has a positive effect on hairy root soybeans under drought treatment.

### 2.8. Change of Drought-, Salt- and Ca^2+^-Responsive Genes in Transgenic Soybean Hairy Roots

To further explore the possible molecular mechanisms involving *GmCBP60A-1* in stress responses, we chose some marker genes based on other reported studies [1,35,36,37,38]. The expression of selected marker genes, including *GmSOS1* [35], *GmAnnexin1* [39,40], *GmCIPK24*, *GmCBL4*, *GmNHX1* [41], *GmCAX3* [42], *GmDREB2*, *GmRD20A*, *GmNAC11*, *GmERD1* [37], *GmMYB118* and *GmMYB174* [43], were upregulating in transgenic hairy roots soybeans under s drought and salt treatments. The upregulation of downstream regulatory genes in the OE hairy roots soybeans was much higher than that of the EV hairy roots soybeans, and upregulation in the RNAi hairy roots soybeans was lower than that in the EV (Figure 9). The data indicated that *GmCBP60A-1* might activate other regulatory genes to enhance soybean tolerance to drought and salt stresses.

## 3. Discussion

The cytoplasmic Ca^2+^ concentration, known as “Ca^2+^ signatures”, is usually induced by a variety of biotic and abiotic stimuli [2]. CaMs/CMLs are important Ca^2+^-sensing proteins that decode the information carried by the Ca^2+^ signatures and transduce the specific Ca^2+^ signal to appropriate effectors [18,19]. Several transcription factor families including CAMTAs, WRKYs, MYBs, NACs and CBP60s have been identified to interact with CaMs/CMLs and play critical roles in plant response to environmental stresses [5,23]. Most CBP60 family members have been found involved in disease resistance, immunity, and SA accumulation [28,44]. However, their functions in response to abiotic stresses remain largely elusive. In this study, 19 members of the soybean *CBP60* genes were identified in soybean genome. And we performed an overall analysis of the *CBP60* genes in soybean, including analysis of their phylogenetic, chromosomal location, gene structure, conserved motifs and expression patterns. Compared to other species, *CBP60* genes in soybean showed a closer relationship with that in *Arabidopsis* because they always clustered closely to each other in the phylogenetic tree (Figure 1).

*Cis-acting* elements are important in the regulation of many processed, including the plant growth and the stress responses [45], and analysis showed that many *cis-acting* elements were related to abiotic stress existed in the promoters of *GmCBP60*s. This result indicated that *GmCBP60*s were related with abiotic stress. To verify our deduction, the promoters of the soybean *GmCBP60*s were analyzed, and found that two important types of *cis-acting* elements existed in their promoters, which were DRE and ABRE. These *cis-acting* elements were reported to involve in plant response to abiotic stress. ABRE *cis-acting* element was related with ABA signal pathway. When the plants were exposed to stress conditions, ABA was accumulated, and which contributed to induction of some stress responsive genes, and ABRE *cis-acting* element played important roles in this induction process [46,47]. Meanwhile, DRE *cis-acting* element was independent of ABA signal pathway and could be bound with DREB transcription factor that was important in regulation of plant tolerance [46,48]. These results further indicated that *GmCBP60*s participated in plant stress response to abiotic stress. In addition, our RT-qPCR result illustrated that some *GmCBP60* genes could be significantly induced under drought and salt stresses, particularly *GmCBP60A-1* which had the highest expression levels under different stress conditions compared with other *GmCBP60* genes (Figure 3). This result elucidated that *GmCBP60A-1* may be very important in the process of plant response to abiotic stress. Further analysis showed that *GmCBP60A-1* and *CBP60g* were divided into C3 subfamily (Figure 1). This result indicated that *GmCBP60A-1* had the high homology with *CBP60g*, which implied that *GmCBP60A-1* may have the same function with *CBP60g*. Meanwhile, in *Arabidopsis*, *CBP60g* have been reported to participate in plant defense responses, and which can contribute to the accumulation of SA [25,26,49,50]. In addition, some studies also revealed that overexpression of *CBP60g* genes in *Arabidopsis* can enhance drought and salt tolerance of transgenic lines [30,31]. These results explained that *GmCBP60A-1* maybe have the similar functions with *CBP60g* gene. To verify our deduction, we generated the *GmCBP60A-1* transgenic plants, and our results showed that overexpression of *GmCBP60A-1* in *Arabidopsis* and soybean could improve plant tolerance to drought and salt stresses (Figure 5 and Figure 6), which further indicated that *GmCBP60A-1* gene participate in plant stress tolerance. In addition, when the plants were treated by severe drought and salt stress, the high level of reactive oxygen species (ROS) was produced in plant, which giving rise to oxidative stress with deterioration of the tissues [51,52,53,54]. The biological function of NADPH oxidase (NOX) in plants is to generate of ROS under normal and stress conditions [51]. Higher level of NOX activity and ROS levels indicates the degree of stress on *GmCBP60A-1*-RNAi plants is more than EV-control, and opposite effects for *GmCBP60A-1*-OE plants under salt and drought stresses (Figure 7G and Figure 8G). Furthermore, MDA is an endogenous genotoxic product of enzymatic and oxygen radical-induced lipid peroxidation and the level of MDA is generally considered as a marker of oxidative stress [53]. It was confirmed that the degree of harm on *GmCBP60A-1*-RNAi plants was more serious than EV-control, and in turn was more serious than *GmCBP60A-1*-OE plants under salt and drought stresses (Figure 7K and Figure 8K). Higher level of proline proved that OE plants had strong tolerance to salt and drought conditions (Figure 7H and Figure 8H). These results also supported that *GmCBP60A-1* was important in the process of plant response to abiotic stress.

Our study showed that *GmCBP60A-1* can trigger the response at the transcriptional levels, such as *GmSOS1*, *GmAnnexin1*, *GmCIPK24*, *GmCBL4*, *GmNHX1*, *GmCAX3*, *GmDREB2*, *GmRD20A*, *GmNAC11*, *GmERD1*, *GmMYB118* and *GmMYB174*. Those genes were found to be upregulated in *GmCBP60A-1*-OE plants under drought and salt conditions (Figure 9). These genes have been employed as salt and drought markers in previously studies [38,55]. For example, GmMYB118 transcription factor might enhance drought and salt tolerance of plants by promoting the expression of stress-associated genes and by regulating flavonoid biosynthesis to reduce ROS content [43,44]. Previous study reported that GmERD1 involved in a cascade of reactions acting directly in response to abiotic stress that plays an important role at an earlier stage in the drought response pathway [37]. In addition, RD20A is considered part of the ABA-dependent pathway [56]. NAC transcription factors are also known to activate the gene expression via stimulating other transcriptional activators such as GmDREB2 or GmERD1 [57]. The GmDREB2 activates transcription by binding to DREs [56,58]. Response to salt-stress, the Ca^2+^-related salt-overly-sensitive (SOS) pathway is triggered [14], including SOS1, SOS2 (CIPK24), SOS3 (CBL4), which is well-known to regulate plant in response to saline conditions [59,60,61,62]. Comprehensive description of the above results suggested GmCBP60A-1 may participate in SOS pathway to enhance the resistance of salinity and in ABA-independent pathway to improve the resistance of drought. The above results suggest that GmCBP60A-1 may play a role in decreasing ROS accumulation and affecting the accumulation of osmoregulation substances via regulation of stress-related gene expression. However, the more detailed mechanisms need to be further studied.

## 4. Materials and Methods

### 4.1. Genomic and Phylogenetic Relationships

CBP60 family members in *Arabidopsis* have been confirmed in previous studies [6,25,26]. The BLASTP program was used to search GmCBP60 protein sequences against soybean genome (*G. max* Wm82.a2. v1) from the EnsemblPlants. We searched the Pfam ID (PF07887) in Pfam website (http://pfam.xfam.org/, accessed on 20 July 2021), and downloaded the related document [63]. Then HMMER v3 and the Hidden Markov Model (HMM) was used to searched for homologous sequences in soybean genome [64,65], then validate them by SMART website (http://smart.embl-heidelberg.de/, accessed on 20 July 2021) [66]. In order to explore the relationship of CBP60 proteins in monocotyledonous and dicotyledonous plants, we obtained the sequences of rice and maize as above described. Finally, the full-length amino acid sequences were aligned using ClustalX [67] under default settings. The resulting alignments in the meg format were submitted to MEGA-X [68] to generate a maximum-likelihood bootstrapped tree (Bootstraps = 1000). The Jones–Taylor–Thoronton (JTT) model was used for the maximum likelihood tree.

### 4.2. Analysis of Gene Structure and cis-Acting Elements

The gene structures of *GmCBP60*s were illustrated using the online program GSDS (http://gsds.cbi.pku.edu.cn/, accessed on 20 July 2021) [69] by comparing predicted coding sequences with their corresponding genomic DNA sequences.

For sequence analysis of GmCBP60 motifs, the online program MEME (http://meme-suite.org/tools/meme, accessed on 20 July 2021) [70] was used for identifying conserved motifs. The resulting map was used to display the distribution of GmCBP60 motifs at the opposite positions in the GmCBP60 proteins using the TBtools v1.075 software [71].

For the analysis of *cis*-elements, DNA sequences of the region 2 kb upstream from the 5’ end of the gene were extracted from the EnsemblPlants database by using TBtools v1.075 software [71]. Next, the potential *cis*-elements of promoters for each gene were analyzed via the PlantCARE database [72], and imaging results were also obtained by the TBtools v1.075 software [71].

For the analysis of conserved domain of GmCBP60 proteins, we obtained the protein sequences from EnsemblPlants and blasted them with the TBtools v1.075 software [71].

### 4.3. Genomic Location of Soybean GmCBP60s

Positional information of *GmCBP60s* on chromosomes of soybean was obtained from the EnsemblPlants database. We used MapGene2Chrom website (http://mg2c.iask.in/mg2c_v2.0/, accessed on 20 July 2021) [73] to obtain the distribution maps of *GmCBP60s* on the chromosomes.

### 4.4. Tissue-Specific Expression Patterns

Transcription databases were obtained from the SoyBase database (https://www.soybase.org/, accessed on 20 July 2021) to analyze the tissue expression patterns of the soybean CBP60 family. TBtools software [71] was used to conduct a visual hierarchical clustering of *GmCBP60s*.

### 4.5. Subcellular Localization of GmCBP60A-1

The CDS of *GmCBP60A-1* (LOC100804548) was amplified without stop codon using gene-specific primer pairs and fused into pJIT16318 vector [74] that contains a CaMV35S promoter and a C-terminal GFP (green fluorescent protein). Approximately 4 × 10^4^ mesophyll protoplasts were isolated from two-week-old *Arabidopsis* leaves and then transfected with the recombinant plasmid (pJIT16318-*GmCBP60A-1*) and the PIP2-mCherry (cytomembrane marker) [75] by PEG-mediated transformation as previously described [74]. Fluorescence in the transformed protoplasts was imaged using a confocal laser scanning microscope (LSM 700, Zeiss, Oberkochen, Germany). The primers of *CBP60A1*-16318-F and *CBP60A1*-16318-R were listed in Appendix A.

### 4.6. Western Blot Analysis

The protein of the plant nucleus, cytoplasm and cytomembrane were isolated by using Minute^TM^ Plasma Membrane Protein Isolation Kit (SM-005, Invent Biotechnologies, Minnesota, America) and then the plant cytomembrane protein was detected by western blot analysis with anti-GFP (HT801-01, TransGen Biotech, Beijing, China) [76]. For the western blot analysis with anti-GFP, TGX Stain-Free FastCast Acrylamide Kit, 12% (Cat #1610185, Bio-Rad, California, America) was used and the protein gel was made following protocol. After electrophoresis, proteins were transferred to PVDF membrane, using a Bio-Rad transfer apparatus. Transfer buffer consisted of 39 mM glycine, 48 mM Tris-HCl, 20% (*v*/*v*) methanol, and 0.037% (*w*/*v*) SDS. Blots were blocked for at least 0.5 h in 5% (*w*/*v*) non-fat dry milk (CAH4859, FUJIFILM, Japan) with PBS buffer. Primary antisera (anti-GFP) were normally diluted in blocking buffer at 1:1000. Antibody incubation was performed for 1 h at room temperature. After that the membrane was washed three times with PBS buffer (five minutes once). The secondary antibody used for immunodetection was a goat anti-mouse conjugated with horseradish peroxidase (HS201-01, TransGen Biotech, Beijing, China), and was diluted at 1:5000 in blocking buffer. Antibody incubation was performed for 1 h at room temperature. After that the membrane was washed three times with PBS buffer (five minutes once). Finally, the proteins on PVDF membrane were detected by enhanced chemiluminescence (Tanon 5200, Shanghai, China) and exposure to x-ray film.

### 4.7. Plant Materials and Growth Conditions

Soybeans “Williams 82” as wild-type soybean in all experiments was grown on moistened soil (vermiculite: humus = 1:1) in a greenhouse with a 14 h light/10 h dark photoperiod, 28/20 °C day/night temperatures, and 60% relative humidity. Growth stages of soybeans are divided into vegetative growth stages (V) and reproductive growth stages (R). Subdivisions of the V stages are designate numerically as V1, V2, V3, through V(n) where (n) represents the number for the last node; except the first two stages, which are designated as VE (emergence) and VC (cotyledon stage) [77].

*Arabidopsis* (*Arabidopsis thaliana*) Columbia-0 (Col-0) ecotype was used in all experiments as wild-type (WT). *Arabidopsis* seeds were surface sterilized, and synchronized at 4 °C for three days [78,79] and germinated in petri dishes containing 1/2 Murashige and Skoog (MS) medium (supplemented with some C-source), followed by transfer to a growth chamber (22 °C, 16 h light/8 h dark photoperiod) to allow germination to start.

### 4.8. RNA Extraction and Reverse-Transcription Quantitative Real-Time PCR (RT-qPCR)

Plant tissues (leaf and root) were sampled after 0, 0.5, 1, 2, 4, 7, and 12 h after treatments initiated. Severe drought stress plants were sampled 0, 4, 12, 24, 36, 48, and 60 h after reaching 40% of soil moisture content.

Total RNA was isolated from 0.1 g (approximate fresh weight) of sampled soybean seedlings using a Fast Plant Total RNA Kit (ZP405-1, Zoman, Beijing, China) and then reversed transcribed into cDNA by a PrimeScript^TM^ RT Reagent Kit (RR037B, TaKaRa, Kyoto, Japan) following the protocol as previously described [74]. The RT-qPCR expression was using TransStart^®^ Top Green qPCR SuperMix (+Dye I) kit (AQ132-11, TransGen Biotech, Beijing, China) and an ABI Prism 7500 sequence detection system (Applied Biosystems, Foster City, CA, USA). The list of the primers used in this study is described in Appendix A. *β-actin* was used as a housekeeping gene [80,81,82] and it is stably expressed in our material (Appendix A). The expression levels were quantified using the 2^−ΔΔCt^ method [83,84]. Results are represented as the mean ± standard deviation (SD) of three technical and three biological replicates.

### 4.9. Stresses Treatments for Genes Expression

Wild-type soybean plants on the V2 stage, were irrigated 15% PEG solution to simulate drought treatment or 150 mM NaCl solution for salt treatment [85,86,87]. Treatments used to evaluate gene expressions at different time points.

Wild-type soybean plants on the V3 stage, were withheld irrigation to induced drought stress. When the soil moisture content was controlled at 40%, samples were taken at different time points to analyze the gene expression patterns. The water content of the soil was monitored throughout the experiment by the gravimetric method [87], which corresponded to the percentage of water in the soil in relation to the dry weight of the soil.

### 4.10. Agrobacterium Tumefaciens Transformation of Arabidopsis

The CDS of *GmCBP60A-1* was amplified with stop codon using gene-specific primer pairs and fused into pCAMBIA-1302 vector that contains a CaMV35S promoter. The recombinant plasmid was introduced in the *A. tumefaciens* strain GV3101 and transformed into WT by floral dip method [88,89] for constructing *GmCBP60A-1* transgenic *Arabidopsis*. The T1 generation seeds were harvested after infection and initially screened by 1/2 MS mediums containing hygromycin (35 mg/L) until the homozygous T3 generation of transgenic *Arabidopsis* lines. As for the seedlings of T1, we extracted the RNA for RT-qPCR to confirm the expression level (Appendix A). The primers of *CBP60A1*-1302-F and *CBP60A1*-1302-R were listed in Appendix A.

### 4.11. In Vitro Germination and Root Growth Assays

Treatments used to evaluate *Arabidopsis* phenotypes in vitro simulated drought and salinity stresses, respectively. The in vitro germination and root growth assays were conducted on 1/2 MS medium containing different concentrations of PEG (3%, 6% and 9%) or NaCl (75 mM and 100 mM) [90,91]. The germination rates were recorded every 12 h until the seeds germinated completely. Three independent biological repeats were performed for seed germination assays. The root lengths of seedlings were evaluated after 7 days of stress treatments [91]. Three biological repeats were performed, and at least twenty seedlings of each genotype were measured by WinRHIZO Pro V2013e software.

### 4.12. A. Rhizogenes Transformation of Soybean Hairy Roots

The CDS of *GmCBP60A-1* was amplified with stop codon using gene-specific primer pairs and fused into pCAMBIA3301 vector under the CaMV35S promoter to generate *GmCBP60A-1*-overexpressing (*GmCBP60A-1*-OE) vector. For the *GmCBP60A-1* RNA interference (*GmCBP60A-1*-RNAi) construct, a specific fragment, including a 200 bp *GmCBP60A-1* sense sequence and its antisense sequence, and a 146 bp intron fragment of maize alcohol dehydrogenase gene as a spacer between the sense and antisense fragments, was synthesized (AUGCT, Beijing, China) and inserted into pCAMBIA3301. Relative sequences were listed in Appendix A. The recombinant vectors and empty pCAMBIA3301 vector (EV) were independently introduced in the *A. rhizogenes* strain K599 and transformed into soybean (*G. max cv.* Williams 82) by the method of *A. rhizogenes*-mediated transformation of soybean [92]. The RT-qPCR analysis of *GmCBP60A-1* expression in *GmCBP60A-1*-OE, EV-control and *GmCBP60A-1*-RNAi transgenic hairy root plants before processing (Appendix A). Each of the hairy root-related experiments was replicated at least three times independently. The primers of *CBP60A1*-3301-F and *CBP60A1*-3301-R were listed in Appendix A.

### 4.13. Salt and Drought Stress Assay

Treatments used to evaluate hairy root soybean plants in vitro were severe drought and salinity. The hairy root soybean plants grew on moistened soil that was weighed to ensure the same amount of water in all the pots. Severe salt stress was induced by irrigating 250 mM NaCl solution at V3 developmental stage [44]. The plants did not receive any additional watering after the single application of the salt treatment. We had been starting to irrigate the plants with 1 L NaCl solution (250 mM) for three times per pot since the soil contained 40% of the water at least, until all plants are gradually death. Severe drought stress was induced by withholding irrigation at V3 developmental stage. The water content of the soil was monitored by the gravimetric method [87] until reach 40% of soil moisture content in relation to soil dry weight. Hairy-rooted soybean plants grew under these conditions and deprived of water until after all plants appeared nearly senesce. All the pots were randomly distributed in a flat (one flat per treatment) and were rotated frequently during the stress periods to minimize effects of the growth environment. In this experiment, five independent transformation hairy root soybean plants were placed in one pot. Six pots formed a group. Each treatment contained three groups *GmCBP60A-1*-OE, EV-control and *GmCBP60A-1*-RNAi transgenic hairy root plants. The same treatment was replicated for three times.

### 4.14. Detection of Physiological Indicators after Salt and Drought Treatments

Contents of proline (PRO-1-Y, Comin, Suzhou, http://www.cominbio.com/, accessed on 20 July 2021, China), MDA (MDA-1-Y, Comin, Suzhou, China), NOX (NOX-1-Y, Comin, Suzhou,China), H_2_O_2_ (BC3590, Solarbio, http://www.solarbio.com/, accessed on 20 July 2021, Beijing, China) and O^2−^ (BC1290, Solarbio, Beijing, China) were measured using their corresponding assay kits following the manufacturer’s protocols. The measurement of chlorophyll content was carried out as described by previously described [93]. All measurements were collected from three biological replicates.

### 4.15. Staining after Salt and Drought Treatments

After drought or salt treatment, we took at least five independent leaves as samples. All samples were sampled from the same location in different hairy root soybean plants. The samples were completely immersed in a 0.5% trypan solution (Solarbio, Beijing, China) for 12 h, then immersed in 75% ethanol for decolorization until the samples turned white, and finally photographed to visualize the staining. Separate sets of five samples were stained with either, 0.5% trypan solution, DAB, or NBT solution for 12, 6, or 4 h, respectively. Samples were subsequently decolorized in 75% ethanol until they turned white. Then all samples were photographed to visualize the staining. All stains were obtained from Solarbio (Beijing, China).

### 4.16. Statistical Analysis

The values were shown as mean ± standard deviation (SD). The data was subjected to Student’s *t* test analysis using statistical soft SPSS 17.0, and the significance (*p* < 0.01) was labeled with **; and *, *p* < 0.05. ANOVA analysis was conducted by SPSS 17.0, and significance (*p* < 0.01) was labeled with different letters.

## 5. Conclusions

We characterized the *GmCBP60A-1* gene as a positive regulator of drought and salt tolerance in soybean. It may enhance drought and salt tolerance by affecting the accumulation of osmoregulation substances, the balance of ROS and the reprogramming of stress tolerance related genes. This study provides new insights in the functional analysis of *GmCBP60* genes in response to abiotic stresses.

## Figures and Tables

**Figure 1 ijms-22-13501-f001:**
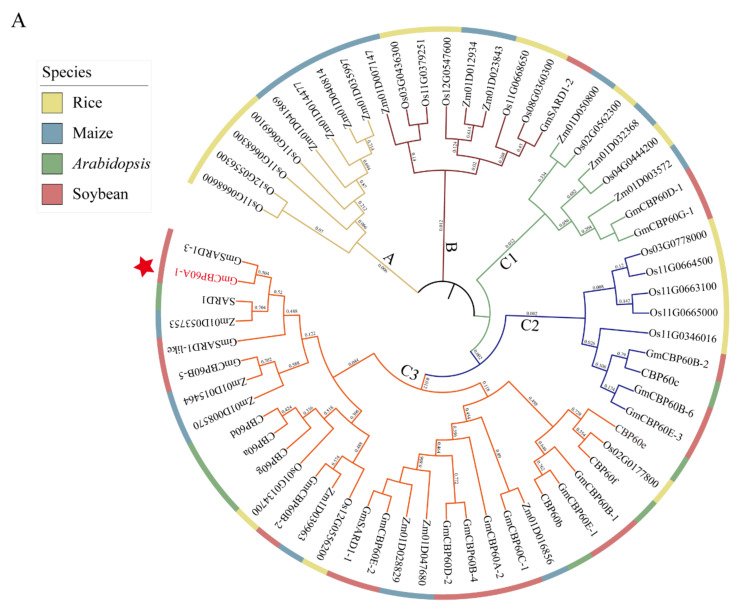
Phylogenetic relationships of CBP60 genes. The unrooted tree was generated with the MEGA-X software using the full-length amino acid sequences of the 63 full-length CBP60 proteins (19, Gm; 19, Os; 17, Zm; 8, At) by the maximum-likelihood (ML) method, with 1000 bootstrap replicates. Different line colors indicate the five major branches. The red star indicates the main study gene in this work. Species acronym used: At, *Arabidopsis*; Gm, soybean; Os, rice; and Zm, maize.

**Figure 2 ijms-22-13501-f002:**
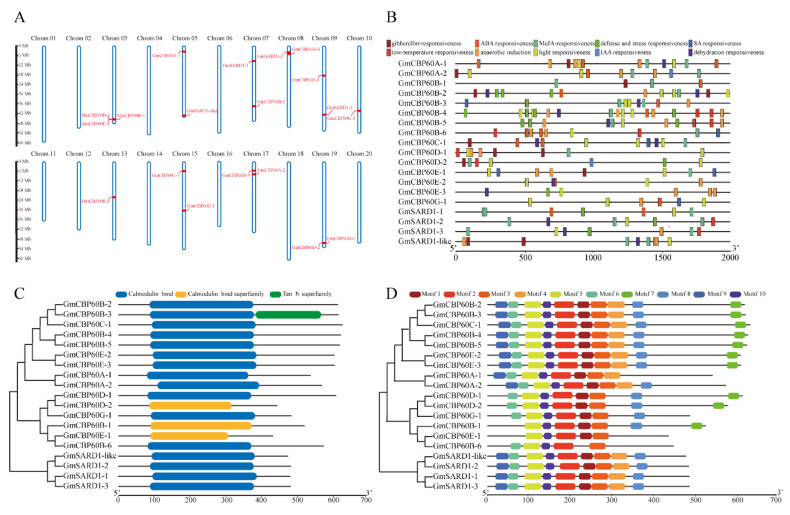
Chromosomal distribution, *cis*-regulatory elements, domains, and motifs of the *GmCBP60* genes. (**A**) The chromosomal distributions of the *GmCBP60* genes in the soybean genome. The red lines point the position of GmCBP60 genes. The chromosome names were set at the top of the chromosomes; (**B**) *Cis*-elements in the GmCBP60 gene promoter regions. Different *cis*-elements were indicated by distinct colored round rectangles; (**C**) Pfam domain present in soybean GmCBP60 family; (**D**) Schematic distributions of the conserved motifs among *GmCBP60* genes. Each colored box represents a motif in the protein.

**Figure 3 ijms-22-13501-f003:**
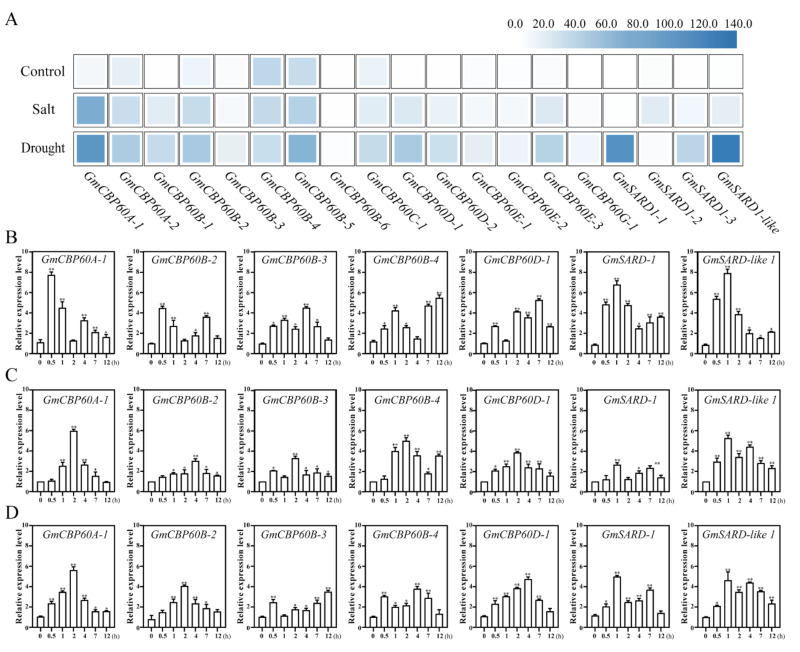
The expression pattern of soybean GmCBP60 genes in response to drought and salt treatment. (**A**) The heatmap of *GmCBP60* genes expression under different treatment in transcriptome [33]. The FPKM values of heatmap are detailed introduction in Appendix A. After salt (**B**), drought (**C**) and PEG (**D**) treatment, expression patterns of seven genes were quantified by RT-qPCR analysis. The soybean actin (*β-actin*) was used as an internal control. The horizontal axis represents time (h), and the vertical axis represents expression level. Three biological replicates were per-formed, and the values are means ± SD.

**Figure 4 ijms-22-13501-f004:**
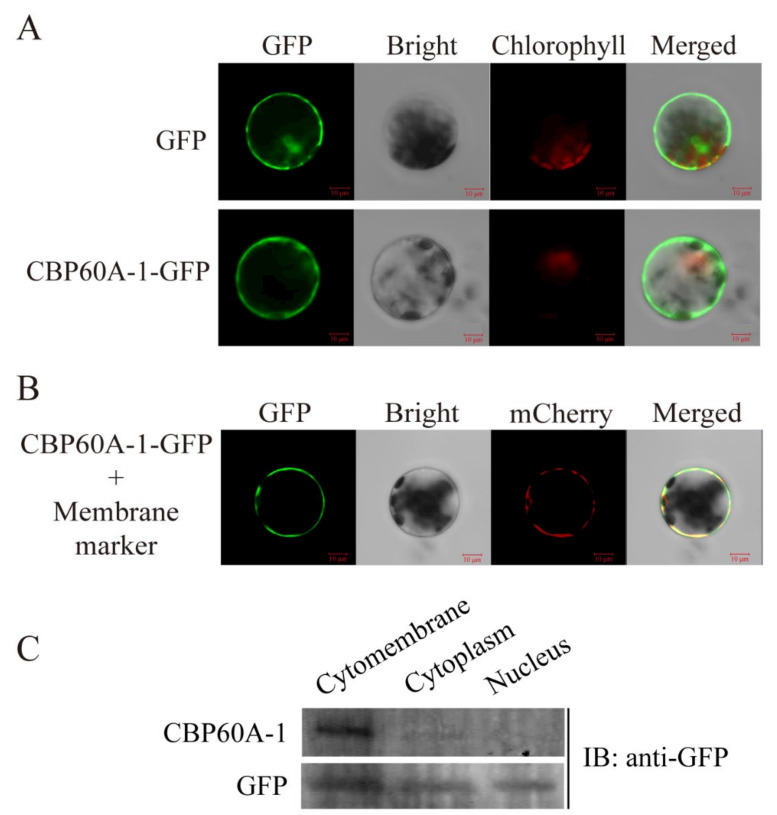
The location of GmCBP60A-1. (**A**,**B**) Green and red fluorescence signals were detected by confocal laser scanning. Scale bars are shown in figure, and the value of scale bar is 10 μM. (**C**) The Western blot result of localization of GmCBP60A-1.

**Figure 5 ijms-22-13501-f005:**
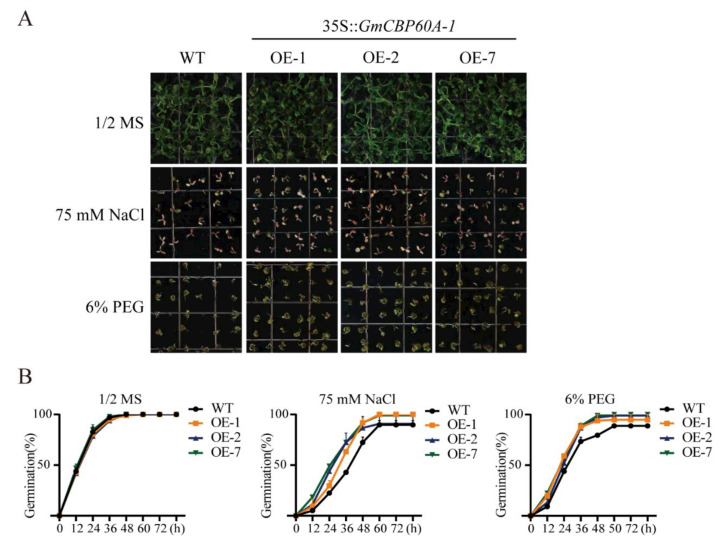
The seed germination of transgenic *Arabidopsis* lines and WT plants under NaCl and PEG treatments and the results of related statistics. (**A**) The seeds of transgenic lines and WT plants were grown in 1/2 MS, 1/2 MS containing 75 mM NaCl and 1/2 MS containing 6% PEG. (**B**) The line chart of germination under different conditions. Three biological replicates were performed, and the values are means ± SD. Values marked with similar letters indicate not statistically significant differences (Student’s *t-*test: *p* ≤ 0.05).

**Figure 6 ijms-22-13501-f006:**
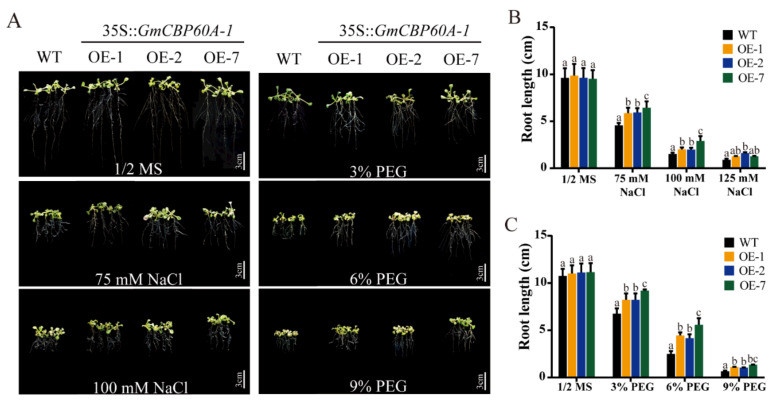
The root growth of transgenic *Arabidopsis* lines and WT plants under NaCl and PEG treatments and the results of related statistics. (**A**) The seedlings of transgenic lines and WT plants were grown in 1/2 MS, 1/2 MS containing different concentrations of NaCl and 1/2 MS containing different concentrations of PEG6000. (**B**) Root growth variation of transgenic lines and WT in the presence of different concentrations of NaCl. (**C**) Root growth variation of transgenic lines and WT in the presence of different concentrations of PEG6000. Three biological replicates were performed, and the values are means ± SD. Values marked with similar letters indicate not statistically significant differences (Student’s *t*-test: *p* ≤ 0.05).

**Figure 7 ijms-22-13501-f007:**
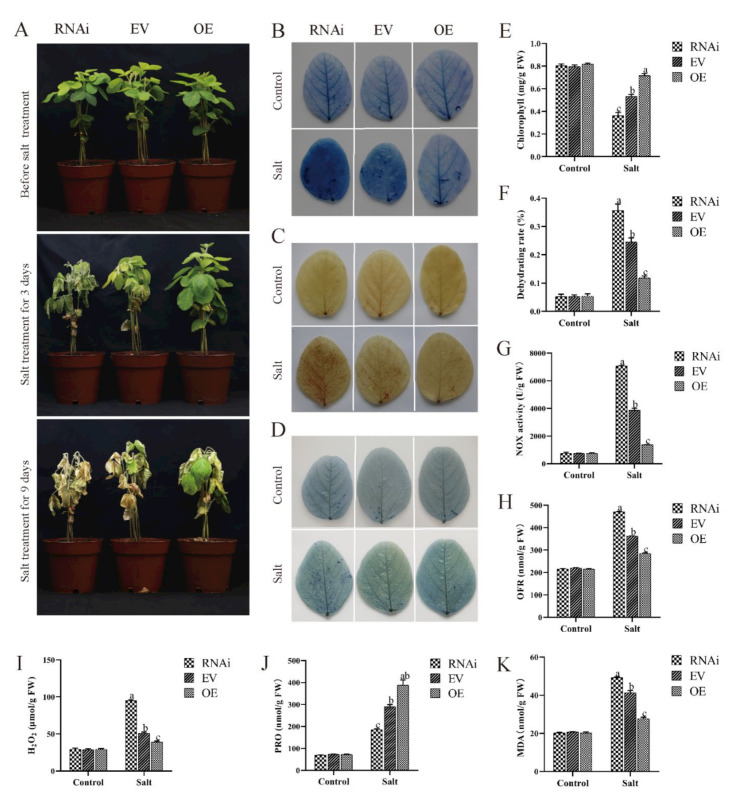
Under salt treatment, the phenotype of hairy root soybean, the states of difference hairy root plant leaves and the results of physiological index. (**A**) Images of salt-resistant phenotypes of the hairy root soybean plants of OE, EV and RNAi from the 250 mM NaCl stress or non-stress treatments are shown. (**B**) Trypan blue staining of dead cells (lives cells do not stain) in the hairy root soybean plants leaves after a week of salt treatment. DAB (**C**) and NBT (**D**) staining of leaves of the hairy root soybean plants of OE, EV and RNAi after salt or no-salt treatment for four days. Chlorophyll content (**E**), water-loss rate (**F**), NOX content (**G**), O^2−^ content (**H**), H_2_O_2_ content (**I**), proline content (**J**), and MDA content (**K**) were detected in leaves of the hairy root soybean plants of OE, EV and RNAi after a week-long salt or no-salt treatment. Three biological replicates were performed, and the values are means ± SD. Values marked with similar letters indicate not statistically significant differences (Student’s *t-*test: *p* ≤ 0.05).

**Figure 8 ijms-22-13501-f008:**
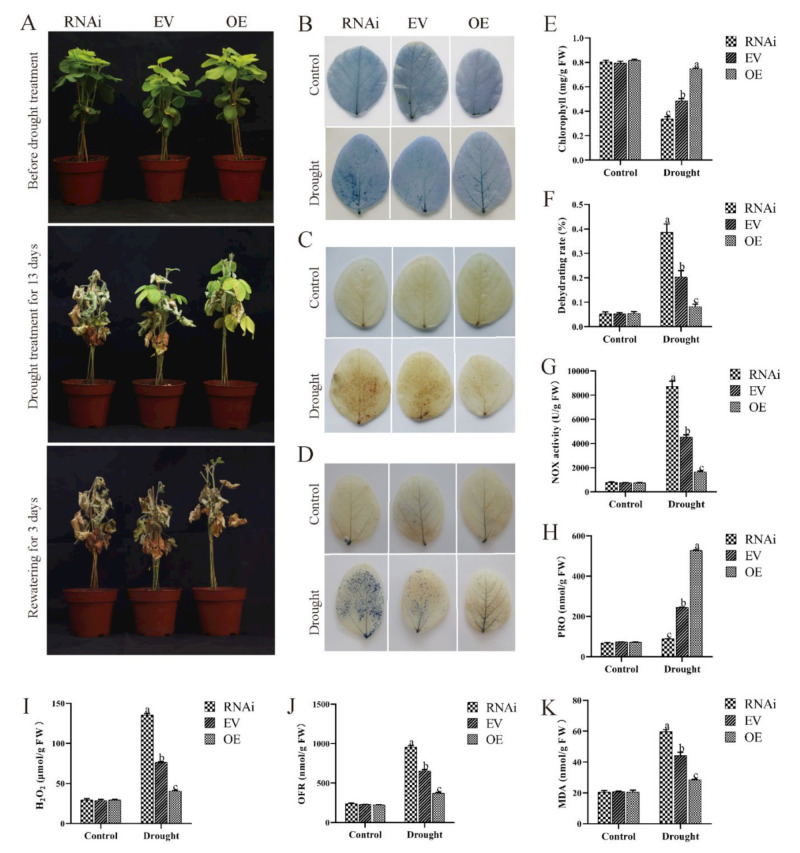
Under drought treatment, the phenotype of hairy root soybean, the states of difference hairy root plant leaves and the results of physiological index. (**A**) the seedlings with 2–5 cm hairy roots were grown for five days in pots under non-drought conditions, and then watering was withheld from plants for 13 days. Survival rates of the water-stressed plants were determined three days after rewatering. (**B**) Trypan blue staining of dead cells (live cells do not stain) of the hairy root soybean plant leaves without irrigation for 11 days. DAB (**C**) and NBT (**D**) staining of the hairy root soybean plant leaves of OE, EV and RNAi after drought or non-drought treatment for a week. The depth of color shows the concentrations of H_2_O_2_ and O^2−^ in the leaves. Chlorophyll content (**E**), water-loss rate (**F**), NOX content (**G**) proline content (**H**), H_2_O_2_ content (**I**), O^2−^ content (**J**), and MDA content (**K**) were detected in leaves of the hairy root soybean plants OE, EV and RNAi under a week-long drought or non-drought treatment. Three biological replicates were performed, and the values are means ± SD. Values marked with similar letters indicate not statistically significant differences (Student’s *t*-test: *p* ≤ 0.05).

**Figure 9 ijms-22-13501-f009:**
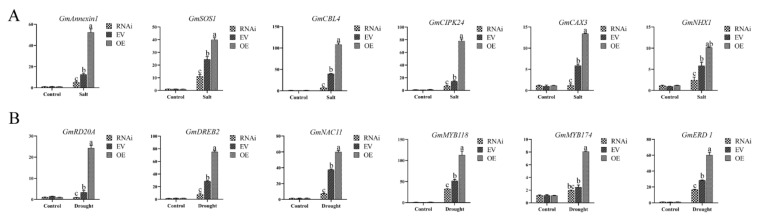
The expression levels of stress-responsive gene in transgenic soybean hairy roots. (**A**) The expression levels of salt-responsive genes under salt stress in transgenic soybean hairy root. (**B**) The expression levels of drought-responsive genes under drought stress in transgenic soybean hairy root. Three biological replicates were performed, and the values are means ± SD. Values marked with similar letters indicate not statistically significant differences (Student’s *t*-test: *p* ≤ 0.05).

**Table 1 ijms-22-13501-t001:** Identify the number of having both ABAR and DER *cis-acting* in the promoter of six GmCBP60 genes.

Gene Name	GmCBP60A-1	GmCBP60B-2	GmCBP60B-3	GmCBP60B-4	GmCBP60C-1	GmCBP60E-2
ABAR	1	1	1	1	2	1
DER	1	2	1	1	2	1

## Data Availability

The datasets presented in this study can be found in online repositories. The names of the repository/repositories and accession number(s) can be found below: NCBI SRA [accession: PRJNA694374].

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
