# Peer review of "Genome-Wide Analysis of the Soybean Calmodulin-Binding Protein 60 Family and Identification of GmCBP60A-1 Responses to Drought and Salt Stresses"

_ijms, 2021, doi:10.3390/ijms222413501_

Round 1
Reviewer 1 Report
The manuscript Qian Yu et. al. (I.D.: ijms-1490617)
shows the possible function of the GmCBP60A-1 in drought and salt stresses.
I have some comment and suggestion for the Authors.
In general, the Authors have to enhance the resolution of figures for better visualization, and increase the size of the letters for better readability of the text in the figures.
When the Authors use the word "membrane", please specify that the membrane is a "plasma membrane".
Line 165, Figure 4: Scale bar is not visible, and please also add values.
Line 202: Please specify what EV means!
Line 447: Authors wrote: “soybean seedlings”. My question: which experiment is based on these samples? Where did the Authors use these samples?
Line 457:
Same as before: Why and in which experiment did the Authors use 15% PEG or 150mM NaCl treatment? The Reviewer found no reference to this experiment in the text.
Line 479: Please add the PEG and NaCl concentration.
Line 485 point 4.12. A. rhizogenes Transformation of Soybean Hairy Roots
The pCAMBIA3301 vector has a 35S promoter followed by the GUS gen with intron. You can find the map of pCAMBIA3301 vector at the following link:
https://www.snapgene.com/resources/plasmid-files/?set=plant_vectors&plasmid=pCAMBIA3301
Why did the Authors clone the CDS of GmCBP60A-1 (LOC100804548) without STOP codon into this vector? Without STOP codon, the Authors generated in-frame or not-in-frame fusion with the GUS gene, which could increase the size of the GmCBP60A-1 protein. This could destabilize or inhibit the protein function.
The Reviewer is experienced in the hairy root transformation, and has some question in this regard.
During the A. rhizogenes transformation, not all, but only the newly formed hairy roots are transformed with the construct of interest.
Please look at the reference 90:
- Kereszt, A.; Li, D.; Indrasumunar, A.; Nguyen, C.D.; Nontachaiyapoom, S.; Kinkema, M.; Gresshoff, P.M. Agrobacterium rhizogenes-mediated transformation of soybean to study root biology. Nature Protocol 2007, 2, 948-952, 791doi:10.1038/nprot.2007.141.
From this Article:
“On average, 25–80% of the hairy roots are co-transformed, that is, carry the T-DNA from both the Ri plasmid and the binary vector used.”
My questions:
1, How did the Authors find/identify/distinguish the non-transformed or co-transformed hairy roots?
In the referenced article, the Authors recommend the followings:
“To facilitate the identification of the transformed roots, use binary or integrative vectors harbouring a reporter gene coding for red or green fluorescent protein”
2, How were the hairy root samples collected for QPCR analysis while avoiding hairy root injury?
3, How many hairy roots were left per plants? If not the same amount, please explain it.
4, In each soybean hairy root experiment, the Authors are required to generate new hairy-root soybean plants transformed by A. rhizogenes. In supplement Figure S4 panel B (line 550 figure legend) has an expression data from the soybean hairy roots.
The Reviewer’s questions: What was the source of these samples? Did the Authors test the gene expression in all newly formed hairy roots before the new experiment/treatment? If not, how could the Authors compare the obtained results from the different experiments? How could the dose-dependent effect be avoided?
Author Response
Dear Professor,
We are grateful for your critical comments and valuable suggestions that have helped us to improve our paper. As indicated in the responses that follow, we have taken all comments and suggestions into account in preparing the revised version of our paper. We edited precision in English language, errors and awkwardly sentences throughout text as possible as we can. All the points revised according to your comments were highlighted as red letters in the revised manuscript. In addition, there are some questions that I sincerely explain to you.
Q 1: When the Authors use the word "membrane", please specify that the membrane is a "plasma membrane".
Response: Dear professor, thanks for your suggestion, I modified it on our manuscript.
Q 2: Line 165, Figure 4: Scale bar is not visible, and please also add values.
Response: Dear professor, scale bars and values are showed on figure 4. In order to avoid this situation, we will enlarge the font of values and scale bars. And other poor-quality pictures were be replaced.
Q 3: Line 202: Please specify what EV means!
Response: Dear professor, the empty pCAMBIA3301 vector were independently introduced in the A. rhizogenes strain K599 and transformed into soybean by the method of A. rhizogenes-mediated transformation of soybean. Seedlings were transformation empty vector, which were called EV seedlings. That explained on line 215-217.
Q 4: Line 447: Authors wrote: “soybean seedlings”. My question: which experiment is based on these samples? Where did the Authors use these samples?
Response: Dear professor, this part described the experiment of RNA Extraction and Reverse-Transcription Quantitative Real-Time PCR (RT-qPCR). According to the different of experiments, we used different tissues of seedlings. The experiments of figure 3B-D took leaves to samples. But the experiments of figure S4B took roots to samples.
Q 5: Line 457: Same as before: Why and in which experiment did the Authors use 15% PEG or 150mM NaCl treatment? The Reviewer found no reference to this experiment in the text.
Response: Dear professor, we added references to this experiment on the manuscript. Drought stress simulated by PEG (15%) had a negative effect on the vegetative growth of all soybean genotypes tested (Sunaryo et al., 1970). 15% PEG treatment and soil drought for osmotic stress; 150 mM NaCl treatment for salt stress (Li et al., 2017). I described the experimental methods in more detail, as follows: When the soybeans plants reached to the V2 stage, they were irrigated 15% PEG solution for simulate drought treatment or 150 mM NaCl solution for salt treatment (Sunaryo et al., 1970;Alvim et al., 2001;Li et al., 2017). And then at the time points of 0, 0.5, 1, 2, 4, 7 and 12 h after treatments were sampled for RT-qPCR assays. Until soybean plants reaching the V3 developmental stage, severe drought stress was induced by withholding irrigation for one-week. The water content of the soil was monitored throughout the experiment by the gravimetric method, which corresponds to the percentage of water in the soil in relation to the dry weight of the soil. After the drought treatment, the first time point was sampled when the soil moisture content was controlled 40%. And then at the time points of 0, 4, 12, 24, 36, 48 and 60 h after treatments were sampled for RT-qPCR assays.
Wright, D., and Lenssen, A.W. (2013). Staging Soybean Development. Agriculture and Environment Extension Publications.
Alvim, F.C., Carolino, S.M.B., Cascardo, J.C.M., Nunes, C.C., Martinez, C.A., Otoni, W.C., and Fontes, E.P.B. (2001). Enhanced accumulation of BiP in transgenic plants confers tolerance to water stress. Plant Physiology 126, 1042-1054.
Li, Y., Chen, Q., Nan, H., Li, X., Lu, S., Zhao, X., Liu, B., Guo, C., Kong, F., and Cao, D. (2017). Overexpression of GmFDL19 enhances tolerance to drought and salt stresses in soybean. PLoS One 12, e0179554.
Sunaryo, W., Widoretno, W., Nurhasanah, N., and Sudarsono, S. (1970). Drought tolerance selection of soybean lines generated from somatic embryogenesis using osmotic stress simulation of polyethylene glycol (PEG). Nusantara Bioscience 8.
Q 6: Line 479: Please add the PEG and NaCl concentration.
Response: Dear professor, thanks for your suggestion, I modified it on line 488- 489
Q 7: Why did the Authors clone the CDS of GmCBP60A-1 (LOC100804548) without STOP codon into this vector? Without STOP codon, the Authors generated in-frame or not-in-frame fusion with the GUS gene, which could increase the size of the GmCBP60A-1 protein. This could destabilize or inhibit the protein function.
Response: Dear professor, we are very grateful for your valuable comments on this, and we apologize for this incorrect description. We modified it to “the CDS of GmCBP60A-1 (LOC100804548) with STOP codon into this vector” on line 483.
Q 8: How did the Authors find/identify/distinguish the non-transformed or co-transformed hairy roots?
Response: Before sampling, we will cut off the root tissues other than the stem nodes, and take the hairy roots at the stem nodes for positive detection.
Q 9: In the referenced article, the Authors recommend the followings: “To facilitate the identification of the transformed roots, use binary or integrative vectors harbouring a reporter gene coding for red or green fluorescent protein”. How were the hairy root samples collected for QPCR analysis while avoiding hairy root injury?
Response: Dear professor, thanks for your suggestion. Firstly, we took the same number and weight of roots from different seedlings. Sampling will be taken when we transplant the seedlings into new pots. In addition, we gave the seedlings a certain recovery time after sampling, not immediately to treat.
Q 10: How many hairy roots were left per plants? If not the same amount, please explain it.
Response: The differences in roots will affect the phenotype of the upper part of the ground. Soybean hairy root seedlings transformed by A. rhizogenes-mediated experiment, there must be a difference in the number of hairy roots possessed by the infected seedlings. But in order to reduce the experimental differences, we will select seedlings with consistent growth to keep the number and volume of new roots of transplanted seedlings basically the same.
Q 11: The Reviewer’s questions: What was the source of these samples? Did the Authors test the gene expression in all newly formed hairy roots before the new experiment/treatment? If not, how could the Authors compare the obtained results from the different experiments? How could the dose-dependent effect be avoided?
Response: Dear professor, the source of these samples was from every hairy root soybean plant transformed by A. rhizogenes. And we tested the gene expression in all newly formed hairy roots before the new experiment/treatment, and we only put a set of results to illustrate that the relative expression level of hairy roots plants is STABLE, all OE is greater than EV and all RNAi is lower than EV which is an effective expression.
Reviewer 2 Report
The article needs corrections. Too many analyzes are described and explained quite chaotically.
Most of the figures are poor quality and need to be replaced (Figure 3B, Figure 9)!
Many abbreviations do not describe what they mean (Line 38 - MARK; Line 123 - ABAR and DER; Line 173: OE and WT; Line 202: OE, EV, and RNAi). The description the authors gave at the end but made it difficult to read.
Line 80: Glycine max must be Italic!
A figure (Figure 3A) is not the result of the authors, but they gave it in the Results section.
"This result indicated ..." - This sentence is repeated many times in section Discussion!
Line 292: "Phylogenetic analysis showed that CBP60 proteins of monocotyledonous and dicotyledonous could be divided into different branches, but branch C contained some maize and rice genes, indicating that these CBP60s may exist before the divergence of monocots and dicots and change with the divergence. " Rewrite it! It is not clear about branch C, the figure showing this is not near the sentences.
Line 390: conversed replace with conserved!
Line 431: "G. max cv. Williams 82" - The newest nomenclature to write names of the cultivars is different!
Author Response
Dear Professor,
We are grateful for your critical comments and valuable suggestions that have helped us to improve our paper. As indicated in the responses that follow, we have taken all comments and suggestions into account in preparing the revised version of our paper. We edited precision in English language, errors and awkwardly sentences throughout text as possible as we can. All the points revised according to your comments were highlighted as red and blue letters in the revised manuscript. And the description of the “Results” section has been greatly revised. In addition, there are some questions that I sincerely explain to you.
Q 1: Most of the figures are poor quality and need to be replaced (Figure 3B, Figure 9)!
Many abbreviations do not describe what they mean (Line 38 - MARK; Line 123 - ABAR and DER; Line 173: OE and WT; Line 202: OE, EV, and RNAi). The description the authors gave at the end but made it difficult to read.
Line 80: Glycine max must be Italic!
Line 390: conversed replace with conserved!
Response: Dear professor, we are sorry to cause you trouble in this regard. We replaced poor quality figures and many abbreviations were added the mean of those on our manuscript. And we modified others, like “Soybean (Glycine max) is one of the most economically and nutritionally crucial crops in the world.” on line 83, and “For the analysis of conserved domain of GmCBP60 proteins, we obtained the protein sequences from EnsemblPlants and blasted them with the TBtools v1.075 software.” on line 406.
Q 2: A figure (Figure 3A) is not the result of the authors, but they gave it in the Results section.
Response: Dear professor, the transcriptome data mentioned in figure 3A of the experimental result belongs to our laboratory and are shared by our laboratory, so I think we can give it in the “Result” section. And this dataset can be found in online repositories. The names of the repository/repositories and accession number(s) can be found below: NCBI SRA [accession: PRJNA694374].
Q 3: Line 292: "Phylogenetic analysis showed that CBP60 proteins of monocotyledonous and dicotyledonous could be divided into different branches, but branch C contained some maize and rice genes, indicating that these CBP60s may exist before the divergence of monocots and dicots and change with the divergence. " Rewrite it! It is not clear about branch C, the figure showing this is not near the sentences.
Response: Dear professor, thanks for your suggestion. We rewrite the result about “Phylogenetic”.
Q 4: Line 431: "G. max cv. Williams 82" - The newest nomenclature to write names of the cultivars is different!
Response: Dear professor, thanks for your suggestion. We motified it to “Soybeans “Williams 82” as wild-type soybean in all experiments was grown on moistened soil (vermiculite: humus = 1: 1) in a greenhouse with a 14 h light/10 h dark photoperiod, 28/20°C day/night temperatures, and 60% relative humidity.” on line 447.
Round 2
Reviewer 1 Report
Dear Authors,
Thank you for your response. The authors corrected all my previous suggestions. I have no more questions or suggestions.